# Hopf Bifurcations in Complex Multiagent Activity: The Signature of Discrete to Rhythmic Behavioral Transitions

**DOI:** 10.3390/brainsci10080536

**Published:** 2020-08-09

**Authors:** Gaurav Patil, Patrick Nalepka, Rachel W. Kallen, Michael J. Richardson

**Affiliations:** Department of Psychology, Centre for Elite Performance, Expertise and Training, Faculty of Medicine, Health and Human Sciences, Macquarie University, Sydney, NSW 2109, Australia; patrick.nalepka@mq.edu.au (P.N.); rachel.kallen@mq.edu.au (R.W.K.); michael.j.richardson@mq.edu.au (M.J.R.)

**Keywords:** multiagent coordination, Hopf-bifurcation, dynamical motor primitives, behavioral dynamics, task dynamics, shepherding dynamics, synergies

## Abstract

Most human actions are composed of two fundamental movement types, discrete and rhythmic movements. These movement types, or primitives, are analogous to the two elemental behaviors of nonlinear dynamical systems, namely, fixed-point and limit cycle behavior, respectively. Furthermore, there is now a growing body of research demonstrating how various human actions and behaviors can be effectively modeled and understood using a small set of low-dimensional, fixed-point and limit cycle dynamical systems (differential equations). Here, we provide an overview of these *dynamical motor*
*primitives* and detail recent research demonstrating how these dynamical primitives can be used to model the task dynamics of complex multiagent behavior. More specifically, we review how a task-dynamic model of multiagent shepherding behavior, composed of rudimentary fixed-point and limit cycle dynamical primitives, can not only effectively model the behavior of cooperating human co-actors, but also reveals how the discovery and intentional use of optimal behavioral coordination during task learning is marked by a spontaneous, self-organized transition between fixed-point and limit cycle dynamics (i.e., via a Hopf bifurcation).

## 1. Introduction

A differentiating factor between novice and expert teams is the development of robust patterns of behavior that enable teams to behave in a responsive and effective manner. The development of such patterns, or coordinative structures [1], within multiagent settings reflects the formation of an interpersonal or multiagent synergy [2]. Typically, the term *synergy* is used to refer to a collection of motor elements (e.g., muscles, joints, limbs, bodies) that are coupled together to act as a single functional unit [3,4]. It is important to appreciate, however, that the formation of synergistic behavior entails two levels of constraints [2,3,4,5]. As illustrated in Figure 1, the first level is structural, whereby neural and biomechanical constraints couple component DoF together into a functional *intra-* or *inter[multi]-person motor synergies*. The second level of constraint is defined during the enactment of a behavioral goal, whereby the specific physical and informational properties of a task and task environment further couple task relevant DoF together to produce an even lower-dimensional pattern of synergistic perceptual-motor behavior [2,4,6,7,8]. 

The perceptual-motor behavior that emerges from the second level of synergistic constraint reflects the task or behavioral dynamics [8,9] that define a given solo and multi-agent activity [8,9,10,11]. Of key significance here is that a growing body of research has revealed that such synergistic perceptual-motor behavior is essentially composed of just two fundamental movement types [12,13]. First, *discrete movements* which are movements that have an unambiguously identifiable start and stop or are bounded by distinct postures, such as when one reaches for an object or target location, taps a key, or throws a ball or a dart [14]—second, *rhythmic movements*, which are movements that are periodic in nature, with the movement pattern recurring at approximately regular time intervals, such as when one waves a hand, pounds a nail in with a hammer, or simply walks. As we detail below, the corresponding implication is that these perceptual-motor behaviors reflect the two elemental behaviors of nonlinear dynamical systems, namely, *fixed-point (discrete)* and *limit cycle (rhythmic)* behaviors. In turn, not only can the task dynamics of complex human perceptual-motor behavior be modelled using these *dynamical primitives*, but by doing so can provide deep insights about the self-organized realization and development of robust and effective behavioral actions.

Our aim here is to detail recent research investigating the task dynamics of multiagent shepherding behavior (two human agents corralling and containing a set of autonomous target “sheep” agents) that provides particularly strong support for the latter claim (e.g., [10,15,16]). Accordingly, we review this recent research, as well as detail how task-dynamic models generated from fixed-point and limit cycle dynamical primitives are not only able to capture the behavioral modes observed in robust individual and multiagent behavior, but also reveal how the discovery and intentional enactment of effective, synergistic oscillator behavior can be spontaneously realized by a nonlinear transition process known as a Hopf bifurcation [17]. Before we do so, however, we first provide an introduction to the formal aspects of the dynamical primitives that form the foundation to modelling and understanding the task dynamics of synergistic human and multiagent perceptual-motor behavior, as well as the Hopf bifurcation process that is entailed within a subset of these dynamical primitives (i.e., limit cycle or self-sustained oscillator systems).

## 2. Dynamical Primitives of Task Actions

As noted above, research on task/behavioral dynamics of goal directed synergistic perceptual-motor behavior has revealed that the majority of human actions are essentially composed of two fundamental movement types, or motor primitives, *discrete* actions (e.g., pressing a button, reaching for a cup) and *rhythmic* actions (e.g., walking, talking), both of which can be integrated together to produce more complex behavioral patterns and sequences (e.g., cursive writing [18], playing a piano [19]).

Historically, research on discrete and rhythmic behaviors has typically been conducted separately. This was simply due, in part, to researchers’ preferential study of one behavioral mode over the other [20]. An unfortunate consequence of this separation of focus, however, is that the functional relationship between discrete and rhythmic movements became a long-standing topic of debate within the motor control literature [13,21]. Whether both types of movements are generated by the same or different fundamental planning mechanisms and how these potential differences affect how these movement types might be mathematically modelled has been of focal interest in this debate [9]. However, despite the debate about the fundamental importance of discrete and rhythmic movements, research has clearly demonstrated that the enactment of discrete and rhythmic motor primitives reflects the behavioral patterns associated with the two fundamental attractors of nonlinear dynamical systems, namely *fixed-point (discrete)* and *limit cycle (rhythmic)* attractors, respectively [9,22,23,24]. Moreover, both simple and complex discrete and rhythmic actions and action sequences can be modelled from these simple *dynamical primitives*. 

Regarding the specifics of the dynamical (differential) equations or functions that can capture these dynamical primitives, several related formulations have been proposed [8,9,20,22,23,25,26,27]. Although the intended application of the different formulations [8,9,22] has resulted in task-specific variations, it is important to note that they all essentially capture discrete (fixed-point attractive) movements using some form of point-goal directed, damped-mass spring system, and rhythmic (limit cycle) movements using either forced (driven) damped-mass spring system or a nonlinear self-sustained oscillator (e.g., Rayleigh, van der Pol, or hybrid oscillators). Accordingly, before reviewing how these dynamical functions can be employed to capture human actions and, in particular, multiagent shepherding behavior, we first briefly introduce the basic formulation and properties of these dynamical primitives.

### 2.1. Fixed-Point Attractor (Damped Mass–Spring) Systems

The simplest example of the kind of damped mass–spring that can be employed to capture the dynamics of discrete movements takes the rudimentary form
(1a)$$m\ddot{x}=-b\dot{x\;}-kx,$$
where *x* represents the position of an object, a body (e.g., an individual’s center of-mass), or an end-effector (e.g., an individual’s hand), $\dot{x\;}$ represents the rate of change of *x* or the first derivative of *x*, which in this case corresponds to the velocity of *x* over time, and $\ddot{x}$ represents the second derivative of *x* or the rate of change of the rate of change of *x* and corresponds to the acceleration of *x* over time. Finally, *m* is the *mass* of the object, *b* is the *damping* (friction) parameter (see Figure 2) that “resists” motion (for *b* > 0) and *k* is a restoring (“spring”) force or *stiffness* that induces motion (for *k* > 0) when the system is not at equilibrium (i.e., when *x* is not at the preferred stable or fixed-point state). 

This mass–spring system (also termed a *simple harmonic oscillator* when *b* = 0) is perhaps one of the most well-known and widely used equations in dynamical systems. Certainly, any student who has ever taken a class on differential equations or dynamical systems will be intimately familiar with this equation. To understand the behavior of this system, let us first set *m* = 1 and, thus, simplify Equation (1a) to
(1b)$$\ddot{x}=-b\dot{x\;}-kx$$

Given that the change of *x* over time is a function of both its current position, *x*, and its current velocity, $\dot{x\;}$, we can also expand Equation (1b) by substituting *x_1_* = *x* and *x_2_* = $\dot{x\;}$ to create a system of two equations, namely
(1c)$$\begin{array}{c} {\dot{x}}_{1}=x_{2}\\ {\dot{x}}_{2}=-bx_{2}–kx_{1} \end{array}$$
where the first equation captures how the position, *x* = $x_{1}$, changes over time and the second equation captures how the velocity, $\dot{x\;}$ = $x_{2}$, changes over time. From Equation (1c), it becomes clear that this mass spring system is a two-dimensional system, in that the behavior of two state variables is being determined to describe the behavior of the system. Illustrating how this system behaves over time therefore requires a 2D plane to graphically represent the systems state at any given time, with one axis corresponding to $x_{1}=x$ and the other axis corresponding to $x_{2}=\dot{x\;}$ (see Figure 2). In dynamical systems terms, this 2D space is the system’s *state* or *phase space* and defines the set of all possible states that the system could adopt. 

Three examples of how the behavior of Equation (1b) can evolve over time for different parameter settings are displayed in Figure 2 using both state space trajectories and time-series graphs. For these examples, *k* = 1, $x_{0}$ = ($\;x_{1}$, $x_{2}\;$) = ( *x*, $\dot{x\;}$) were set to arbitrary initial conditions and the damping parameter, *b*, was set to 0.5, 0, and −0.5 from left to right, respectively. The behavior exhibited in the bottom left of Figure 2 corresponds to the terrestrial scenario in which the position of the system being modelled (i.e., object, body, or end-effector) is pulled away from its resting position at *x* = 0, and then oscillates back and forth at a smaller and smaller amplitude until it eventually returns to *x* = 0 and comes to a stop. In this case, the parameter *b* acts like a friction force, damping out the potential energy that results from extending or compressing the spring or stiffness force, with the state *x** = (*x,*
$\dot{x\;}$) = (0, 0) corresponding to a stable fixed-point attractor. The opposite behavior is observed in the bottom right of Figure 2, where *b* = −0.5. Here, the negative value of *b* results in a kind of a negative-friction force, with the system oscillating away from *x** = (*x,*
$\dot{x\;}$) = (0, 0). Hence, the state *x^*^* corresponds to an unstable fixed-point repeller. Finally, when *b* = 0 (bottom middle of Figure 2), the system simply oscillates harmonically at an amplitude equal to the initial position of the mass when released (i.e., the value of $x_{0}$). In this case, *x** is neither stable nor unstable and is therefore considered to be neutrally stable.

To fully appreciate how this simple mass–spring system can be employed to capture discrete motor actions, let us expand Equation (1b) to the form
(2)$$\ddot{x}=-b\dot{x\;}-k\left(x-x_{des}\right)$$
where the new term $x_{des}$ specifies the preferred position or goal state for the system. Moreover, let the system be critically damped (or overdamped) such that, when *x* is moved away from $x_{des}$, the system returns to $x_{des}$ without any oscillation. This is illustrated in Figure 3 (for *k* = 1, $x_{0}$ = ($\;x_{1}$, $x_{2}\;$) = ( *x*, $\dot{x}\;$) ) set to an arbitrary initial condition, $x_{des}$ set to an arbitrary desired value and the damping parameter, *b*, set to 1.8), such that we can use this simple goal-directed mass–spring systems (i.e., Equation (2)) to model human end-effector trajectories, including, for example, the movement trajectories exhibited by human agents during route navigation tasks [28], as well as during reaching and object movement tasks (e.g., [9,29]).

### 2.2. Limit Cycle Systems

A limit cycle is a closed orbit in state space that a system’s trajectory converges towards over time such that, after a time, the system’s behavior is fixed along that orbit (i.e., it is limited to cycle about that orbit). This kind of attractor is characteristic of periodic or oscillatory behaviors that exhibit a stable spatial-temporal pattern over time (e.g., cycle with a stable frequency and amplitude over time) and, moreover, return to that stable spatial-temporal pattern when perturbed.

There are several differential systems that are characterized by limit cycle attractors. The most well-known are self-sustained oscillator systems and the most well-known of these is the *van der Pol oscillator*, which can take the form
(3a)$$\ddot{x}=-b\dot{x}-cx^{2}\dot{x}-kx$$
where x, $\dot{x},$ and $\ddot{x}$ correspond to the position, velocity, and acceleration of a point mass object (with mass = 1), respectively. As in Equations (1a)–(1c), *k* is a stiffness parameter and *b* the linear damping parameter. The function $cx^{2}\dot{x}$ is the van der Pol term which corresponds to a nonlinear dissipative function. As was the case with Equation (1b), we can substitute $x_{1}$ = *x* and $x_{2}$ = $\dot{x\;}$ to split Equation (3a) into a system of two equations:(3b)$$\begin{array}{c} {\dot{x}}_{1}=x_{2}\\ {\dot{x}}_{2}=-bx_{2}-cx_{1}^{2}{\dot{x}}_{2}-kx_{1}, \end{array}$$
such that the behavior of this system is also best visualized in a 2D state space. Example state space trajectories are displayed in Figure 4, using a range of different initial conditions when *b* and *k* = 1, and c = 0.5 and 1. As can be discerned from Figure 4, for any initial state, the system’s trajectories always converge towards and moves around a specific closed orbit, the limit cycle attractor. That is, if the system’s initial state lies inside the orbit of the limit cycle, the state of the system gradually spirals out towards the stable limit cycle trajectory. Similarly, if the system’s initial state lies outside the orbit of the limit cycle, the system spirals in towards the stable limit cycle trajectory.

A more complex dynamical system that exhibits limit cycle dynamics and one that has been widely used to model human rhythmic limb movements is
(4)$$\ddot{x}=-b\dot{x}-w{\dot{x}}^{3}-cx^{2}\dot{x}-kx,$$
where b, c, and k are the same damping and stiffness parameters in Equation (3a) and [$w{\dot{x}}^{3}$] is a second nonlinear Rayleigh escapement term. The nonlinear oscillator in Equation (4) is called the hybrid oscillator (Kay et al., 1987) because it contains nonlinear damping terms from both the van der Pol [$cx^{2}\dot{x}$] [30] and Rayleigh [$w{\dot{x}}^{3}$] [31] oscillators. This system played a key role in the early development of the *coordination dynamics* approach to rhythmic motor behavior and coordination [32,33] as it better captured key features of human rhythmic limb movements compared to the van der Pol and Rayleigh oscillators alone. Specifically, the Hybrid oscillator is able to replicate the amplitude–frequency relationships observed in human oscillatory movement [32].

### 2.3. Hopf Bifurcation: From Discrete to Rhythmic Behavior

It is important to appreciate that nonlinear self-sustained oscillations (e.g., Equations (3a) and (4)) can exhibit both fixed-point and limit cycle behavior depending on the value of *b*, the linear damping coefficient, with *b* = 0 corresponding to the critical point at which this behavioral transition occurs. That is, when b>0 , Equations (3a) and (4) exhibit fixed-point dynamics, with the systems returning to their equilibrium state (*x* = 0) when perturbed. However, when *b* < 0, the equilibrium point (*x* = 0) becomes unstable and the system begins to exhibit limit cycle behavior. This bifurcation or transition from a fixed-point to limit cycle behavior is called a Hopf bifurcation [17]. 

As an illustration of a Hopf bifurcation, let us again consider the van der Pol oscillator similar to that from Equation (3a) where the damping terms are combined (i.e., μ = *c* = −*b)*:(5)$$\ddot{x}=-\mu \left(x^{2}-1\right)\dot{x}-kx$$
such that, when μ<0, the system is positively damped and, thus, exhibits fixed-point dynamics akin to that illustrated in Figure 2 and Figure 3. When μ>0, however, the system exhibits limit cycle behavior, akin to that in Figure 4. Thus, as illustrated in the bifurcation diagram in Figure 5, when the value μ is scaled up or down, Equation (5) spontaneous transitions between fixed-point and limit cycle behavior. In short, a Hopf Bifurcation occurs. 

Hopf bifurcations are ubiquitous in nature and reflect the general dynamical principle by which rhythmic activity emerges in physical and biological systems [34,35,36]. With regard to Hopf bifurcations in human behavior more specifically, transitions between discrete and rhythmic behavior have been explored in rhythmic aiming tasks which require the movement of the hand between two different targets [37,38,39]. Within this task paradigm, spontaneous transitions from fixed-point to limit cycle behavior can emerge to facilitate the ability of individuals to adhere to short timing requirements [27]. Alternatively, rhythmic control can break down to discrete actions when accuracy is paramount [39], or if the movement is slow [40]. In limb coordination tasks, Hopf bifurcations and the spontaneous emergence of limit cycle behavior are known to occur to maintain unstable coordination patterns [35] and ensure the stable coordination of limb movements with visual stimuli moving in orthogonal directions [41]. These transitions can also occur between trials of a given task, such as the formation of a rhythmical pattern of task initiation onsets during a repetitive object throwing task [21]. 

### 2.4. Dynamical Perceptual-Motor Primitives in Individual Behavior

Appreciating the importance of fixed-point and limit cycle functions for modelling human motor behavior, Schaal and Ijspeert and colleagues [9,23], have referred to these dynamical processes as *dynamical motor primitives***.** However, it is worth pointing out that, because the above mentioned dynamical primitives can only ever be functionally defined and modulated to capture goal-directed task behavior when perceptually coupled or tuned to task relevant physical and informational properties of the environment (e.g., object goal locations, stimulus rhythms, other agents), as in Equation (2) for example, the phrase *dynamical perceptual-motor primitive* (DPMP) is perhaps more appropriate, and will be employed here. 

The importance of DPMPs is that they provide a highly generative set of dynamical functions for developing low-dimensional models of synergistic human perceptual-motor behavior. Indeed, the general DPMP hypothesis is that given the right generative formulation, the spatiotemporal patterning of all human end-effector or joint-limb perceptual-motor behavior can be modelled using a DPMP. With regard to the pragmatics of modelling perceptual-motor behavior, relying on DPMPs also significantly reduces the complicated, trial-and-error system identification problem to that of identifying and parametrizing task relevant forcing and coupling functions. 

Motivated by this understanding, a wide range of research over the last 30 years has effectively modeled human perceptual-motor behaviors using DPMPs. For instance, Ref. [9] demonstrated how simple reaching, rhythmic wiping, and cranking tasks can all be modelled using simple task-specific systems of fixed-point attractors and limit cycle attractors acting on corresponding *end-effectors* (e.g., hands for reaching) linked to an object/task-oriented reference frame [42]. Furthermore, Sternad and colleagues have demonstrated that stable oscillations can be modelled by limit cycle dynamics and directly attributed to the oscillatory nature of the signals in the CNS [24,25,26,43]. In particular, Ref. [25] laid the groundwork for combining multiple dynamic primitives, which can interact with each other in order to generate complex observable motion. More recently, Ijspeert and colleagues [23] provided a generic modeling framework for multidimensional systems by modeling each dimension of action or movement as a combination of separate DPMPs. 

At this point, it is worth reiterating that, because human movements and actions are enacted in the pursuit of some task goal, DPMPs must be coupled to task-relevant properties of the environment to effectively model human perceptual-motor behavior. As referenced throughout this paper so far, this is perhaps best detailed in Saltzman and Kelso’s [9] *Task Dynamics* approach, and more recently in Warren’s [8] *Behavioral Dynamics* approach, with the latter approach also emphasizing how a modelling task directed perceptual-motor behavior involves identifying control laws that minimize some task quantity of the agent–environment system. For example, reaching for an object involves the minimization of one’s hand to the target location [9], or the deceleration of a vehicle to avoid collision involves maintaining the control variable *tau* under a specific value [44,45]. Accordingly, modelling the behavioral dynamics of individuals has been particularly successful with regard to easily quantifiable task objectives, such as reaching and catching [46], walking through a cluttered environment [47], and juggling [48]. However, the same DPMP based task/behavioral dynamics approach can also be extended to model the dynamics that unfold during complex (multi)agent-environment task contexts, such as when two or more individuals are sorting and passing objects [29], intentionally or unintentionally synchronizing their limb or body movements (e.g., [49,50,51]); avoid colliding into each other [52], or are moving or navigating together within a crowd [53].

## 3. Hopf Bifurcation in Multiagent Activity: A Cooperative Shepherding Example

The recent work of Nalepka and colleagues [15,16,54,55] on multiagent shepherding provides an excellent example of how a DPMP based task/behavioral dynamics approach can provide a rich understating of individual and multiagent perceptual-motor behavior. In these experiments, the researchers investigated the emergence of coordinated perceptual-motor strategies in a complex task environment that required two human-actors to corral and contain a set (herd) of evasive target agents. The shepherding task, derived from the natural phenomenon of herding of sheep by dogs [56], is presented to participants as a video game. Participants stand on either side of a game field and control ‘herding agents’ (HAs) using handheld motion sensors or touch screen pens. The task of the participants is to control the HAs to successfully corral and contain the set of ‘target agents’ (TAs), typically ranging from 3 to 7 targets, within a red containment region located on the game field. When left unperturbed, TAs exhibit Brownian motion, and thus naturally disperse if left alone. Importantly, however, the TAs are repelled away from the participant controlled HAs, such that, when an HA is within a critical distance from a TA, the TA flees in the opposite direction. Thus, active actions by both participants are required to corral and contain the TAs within the containment region. Task trials are typically between 1 to 2 min, with a trial deemed successful if a participant dyad is able to contain the TAs within the containment area for a specified period or percentage of trial time (e.g., 70% of a 1-min trial). 

Dyads quickly learned that, to prevent the TAs from escaping the game field, or dispersing too far from the containment region, they need to select and recover the TA that is farthest from the containment region. Additionally, due to the placement of participants on either side of a table, participants typically subdivided the task space so that only TAs on the participant’s side of the game field are most often pursued. This behavioral strategy was referred to as search and recover (S&R) behavior and was a viable strategy to contain the TA herd within the containment region—especially when the number of TAs is low (i.e., <4) (see Figure 6). However, when the task required the containment of larger herds (i.e., 4 to 7), S&R behavior is difficult to maintain and does not lead to successful task performance. Indeed, as TA herd size increases, many dyads fail to find a solution to the shepherding task within a standard 45 to 75-min experimental session. For dyads who succeeded, however, a qualitatively distinct mode of behavior emerges. Instead of selecting and pursuing individual TAs that move away from the containment region, or from the rest of the herd, successful dyads in more difficult task conditions discovered a more effective solution involving oscillatory movements that encircled the entire herd. In this way, repulsive forces become equally distributed amongst all TAs that led to the formation of a centrally fixed “clump” (see Figure 6). Given the dyadic nature of the task, this solution is most effectively actualized when both participants perform these oscillatory movements on their respective sides of the field. Moreover, consistent with research on intra- and inter-personal rhythmic coordination (e.g., [33,49,57,58]), dyads also tend to be attracted towards performing oscillatory movements in an in-phase (0°) or anti-phase (180°) manner. Accordingly, this behavioral mode was referred to as coupled oscillatory containment (COC).

An important thing to note about COC behavior is that it is only employed when the herd is corralled sufficiently close to or within the containment area. That is, dyads that discover and employ COC behavior to contain a TA herd still employ S&R behavior to first corral the TAs when widely dispersed. Therefore, dyads who discovered COC behavior also learn when it was appropriate to transition between S&R and COC behaviors.

Another feature of dyads who discovered COC behavior is understanding that encirclement behaviors in general can be used to keep the TA herd contained. Although in the original research this was actualized by both participants producing semi-circular movements around the herd, encircling by just one of the participants is a readily perceptible and is often actualizable by expert herders [55]. For instance, participants readily transition from COC behavior to one participant performing circling movements around the entire TA herd when their partner must pursue a newly introduced TA or when a TA is perturbed outside a contained herd [55].

Perhaps the most interesting aspect of COC emergence, however, is that both its discovery and actualization occur suddenly. Anecdotally, participants who discovered COC behavior describe this discovery as a moment of cognitive insight—a ‘*eureka!*’ moment. In the learning sciences, such moments of cognitive insight can be understood as a cognitive reorganization of the problem itself, which has been operationalized as a nonlinear cognitive phase transition or bifurcation [59]. This also seems to be the case in the shepherding task, in that both the discovery and subsequent initialization of COC behavior reflects a nonlinear, behavioral mode bifurcation.

### 3.1. The Task-Dynamic Model for Multiagent Shepherding

The behavioral modes adopted by dyads can be elucidated by creating a DPMP based task/behavioral dynamic model of the shepherding problem. As with task/behavioral dynamics models in general [8,9], this model defines the dynamics in terms of the task-relevant state variables necessary for task completion. For example, the successful capture of a fleeing TA requires the minimization of the difference between the HA and TA’s positions, embodied by the participant via the movements of the relevant effector (i.e., the hand holding the motion controller).

For the multiagent shepherding task, the containment region is defined as the task’s goal location for which all TAs must be corralled towards. Thus, the goal for participants is to minimize the distance of the TAs from the containment region. For modeling convenience, the containment region can be defined as the origin (0,0) in the task’s coordinate space. Using a polar coordinate system, the task is then considered completed when the radial position of all TAs is within the containment boundary, $r_{\Delta }$. The HA, then, needs to control both the radial and angular components of their position to move and repel the TAs towards the containment region. See Figure 7 for a depiction of the corresponding shepherding model task space and key model variables and parameters.

To effectively capture the behavioral dynamics exhibited by participant dyads completing the shepherding task, the proposed model need to capture the following: (i) S&R; and (ii) COC behavior; as well as the potential for (iii) stable in-phase and anti-phase coordination to emerge during COC behavior; and finally (iv) include the potential for nonlinear transitions back-and-forth between S&R and COC behavior.

In order to achieve this, Nalepka and colleagues [10,15,54] first defined the critical task of HAs as moving towards and corralling TAs whose radial distance was farthest from the containment region. Recall that S&R entailed herders (participants) discretely moving to and corralling the farthest target(s) on their side of the task field and, thus, the radial distance ($r_{i}$) of HA-*i* (where *i = 1* or *2* for the dyadic shepherding task) during S&R behavior reflected point attractive dynamics. Thus, Nalepka and colleagues were able to model each HA’s radial distance, $r_{i},$ using the following environmental coupled mass–spring systems,
(6)$${\ddot{r}}_{i}=-\alpha_{r}{\dot{r}}_{i}-\omega_{\theta }^{2}\left(r_{i}-(r_{T,i}+r_{min}\right))$$
which is similar to Equation (2) and where $\dot{r}$ and $\ddot{r}$ correspond to the velocity and acceleration of HA-*i*’s radial distance, respectively, $r_{T,i}$ is the radial distance of the farthest TA on HA-*i*’s side of the game field, and $r_{min}$ is a fixed parameter that specifies HA-*i*’s minimum preferred radial distance from a TA during shepherding to ensure repulsion towards the goal. In short, Equation (6) operates to reduce the radial distance of HA-*i* with respect to the radial distance of the currently pursued TA, $r_{T,i}$ (i.e., the farthest TA from the containment region HA-*i*’s side of the task field). The coefficients $\alpha_{r}$ and $\omega_{\theta }^{2}$ correspond to the dampness and stiffness parameters, which vary the rate with which *r* converges to $r_{T,i}$.

During S&R behavior, a HA’s radial angle ($\theta_{i}$) must also exhibit fixed-point dynamics similar to that defined in Equation (6), such that each HA-*i*’s radial angle $\theta_{i}$ is attracted toward the radial of angle of the TA ($\theta_{T})$ farthest from the containment region. However, the function that defines the dynamics of $\theta_{i}$ for each HA also needs to entail the potential for oscillatory or limit cycle dynamics for COC to emerge. As detailed above, for human limb movements, this is best captured using the Hybrid nonlinear self-sustained oscillator (i.e., Equation (4)). Accordingly, Nalepka and colleagues captured the dynamics of an HA’s radial angle using
(7a)$${\ddot{\theta }}_{i}=-\alpha_{\theta }{\dot{\theta }}_{i}-\beta {\dot{\theta }}_{i}^{3}-\gamma \theta_{i}{\dot{\theta }}_{i}-\omega_{\theta }^{2}\left(\theta_{i}-\theta_{T}\right)$$
where and ${\dot{\theta }}_{i}$ and ${\ddot{\theta }}_{i}$ correspond to velocity and acceleration of radial angle, respectively, $\alpha_{r}$ and $\omega_{\theta }^{2}$ correspond to the dampness and stiffness parameters, and $\beta {\dot{\theta }}_{i}^{3}$ and $\gamma \theta_{i}{\dot{\theta }}_{i}$ are the nonlinear Rayleigh and van der Pol terms. 

When $\alpha_{\theta }$ > 0, Equation (7a) mirrors the fixed-point dynamics as in Equation (6), such that the HA’s angular position ($\theta_{i}$) and velocity (${\dot{\theta }}_{i}$) is attracted towards and converges upon the radial angle of the pursued TA, $\theta_{T}$. Collectively then, when $\alpha_{r}$ > 0 and $\alpha_{\theta }$ > 0 in Equations (6) and (7a), respectively, the system (modeled HA) moves towards the radial position ($r_{T}$, $\theta_{T}$) of the pursued TA. This results in robust S&R behavior and although the switch between pursued TAs (i.e., what is specified as the farthest TA [$r_{T}$
$\theta_{T}]$) leads to discontinuous changes in $r_{T}$ and $\theta_{T}$, the resultant model still results in continuous changes in behavior. That is, the modeled HA seamlessly moves between and corrals farthest TA to farthest-TA as the radial distance and angle ($r_{T}\;\theta_{T})$ of the TA farthest form the containment area changes and fluctuates changes over time.

Importantly, when $\alpha_{\theta }$ < 0, Equation (7) results in limit cycle or oscillatory behavior at a frequency approximately $\frac{\omega }{2\pi }$ Hz, centered around $\theta_{T}$, with an amplitude equal to $2\sqrt{\frac{\left|\alpha_{\theta }\right|}{\gamma }}$. Again, the inclusion of both the Rayleigh ($\beta {\dot{\theta }}_{i}^{3}$) and van der Pol ($\gamma \theta_{i}{\dot{\theta }}_{i}$) terms results in angular dynamics that exhibits the amplitude–frequency and peak velocity–frequency relationship exhibited by human actors [32]. Additionally, by extending Equation (7a) such that the radial angles of two HAs are coupled (HA-*i* to HA-*j*) in the following way:(7b)$${\ddot{\theta }}_{i}=-\alpha_{\theta }{\dot{\theta }}_{i}-\beta {\dot{\theta }}_{i}^{3}-\gamma \theta_{i}{\dot{\theta }}_{i}-\omega_{\theta }^{2}\left(\theta_{i}-\theta_{T}\right)+\left({\dot{\theta }}_{i}-{\dot{\theta }}_{j}\right)\;(A+B{\left(\theta_{i}-\theta_{j}\right)}^{2})$$
where the coupling function, $\left({\dot{\theta }}_{i}-{\dot{\theta }}_{j}\right)\;(A+B{\left(\theta_{i}-\theta_{j}\right)}^{2})$, creates the potential for both in-phase (0°) and anti-phase (180°) patterns of coordination to occur (see Haken et al., 1985; for more details). Parameters A and B index the strength of the coupling, such that when |4B| > |A| stable in-phase and anti-phase can both emerge. Accordingly, by combining Equations (6) and (7b), Nalepka and colleagues were able to generate an HA model that was not only able to produce stable S&R behavior (when $\alpha_{\theta }$ > 0), but also robust in-phase and anti-phase COC behavior (when $\alpha_{\theta }$ < 0). 

At this point, it is worth emphasizing that the above model contains nothing more than the basic DPMP functions outlined earlier—and that, by simply coupling the radial angle and distance functions to task relevant environmental properties (i.e., farthest TA and co-Herder), the above systems are able to effectively capture all of the behavioral modes exhibited by human actors completing the multiagent shepherding task; i.e., model requirements, (i), (ii), (iii) listed above. Moreover, by simply switching the sign of $\alpha_{\theta }$ from positive to negative, a Hopf Bifurcation occurs in Equation (7), such that the system can spontaneously transition from S&R to COC behavior (and back again). Following this realization, Nalepka and colleagues were then also able to capture the context-sensitive switching between S&R and COC behavior (model requirement (iv) above) via a parametric control law that define the value of $\alpha_{\theta }$ with respect to the task goal (i.e., containing TAs within the containment area). More specifically, Nalepka and colleagues postulated the following control law to induce spontaneous transitions between S&R and COC behavior,
(8)$${\dot{\alpha }}_{\theta }=-\delta \left(\alpha_{\theta }-\varepsilon \left(r_{T}-r_{\Delta }\right)\right)$$
where $r_{\Delta }$ is the critical boundary distance for the farthest TA ($r_{T}$) to be considered contained with the TA herd, and the parameters δ and ε operate to control the rate at which $\alpha_{\theta }$ changes. Briefly stated, when $r_{T}$ > $r_{\Delta }$, the farthest TA or the TAs in general are not defined (perceived) as contained as a herd and $\alpha_{\theta }$ becomes positive resulting in Equation (7) exhibiting fixed-point behavior and, thus, the collective systems (Equations (6), (7b) and (8)) exhibit S&R behavior. In contrast, when $r_{T}$ < $r_{\Delta }$, the farthest TA or the TAs in general are defined (perceived) as being contained as a herd, such that $\alpha_{\theta }$ becomes negative and a Hopf bifurcation occurs, resulting in coupled rhythmic, limit cycle behavior of COC. 

To test the validity of the above shepherding model, Ref. [16] recently embodied the above shepherding model (i.e., the systems of Equations (6), (7b), and (8)) into the control architecture of a virtual artificial player that was designed to complete the multiagent shepherding with human novices in a virtual reality environment. As expected, the artificial player not only produced and successfully transitioned between S&R and COC behaviors in real-time, but exhibited both in-phase and anti-phase patterns of COC behavior with the human novice co-actors. Furthermore, most participants were not able to discern that their virtual co-actor was an artificial (model controlled) player and thought they were completing the task with another participant remotely. Thus, not only was the model able to reproduce the behavioral dynamics exhibited by human actors, but it was able to do so in a way that seemed human-like, indicating that the model, including the appropriate coupling function (see also [60]), captured the essential features of human task behavior.

### 3.2. Hopf Bifurcations as a Signature of Intentional Dynamics

Recall that not all participants who perform the shepherding task discover or employ COC behavior and therefore do not learn when it is appropriate to transition between S&R and COC behavior. This implies that something like the hypothetical control law specified in Equation (8) needs to be learned by participants. Importantly, however, information specifying COC as a potential and effective solution to TA containment can emerge during S&R behavior and, when it does, appears to ultimately lead to its discovery. More specifically, a recent analysis and simulation work (Auletta et al., in prep and [61]) suggests that, when the TA herd is sufficiently clumped together (i.e., when all TAs are within a critical distance, $r_{\Delta },$ and begin to form a single herd), implicit oscillatory behavior is often induced within the movements of participants (HAs) (see Figure 6). Such induced oscillatory behavior seems to occur prior to the full discovery and intentional implementation of COC behavior. Thus, consistent with a more complex dynamical systems account of human behavior, these inter-participant (HA) and TA interactions during ongoing task performance appear to scaffold the creation of intentional dynamical behavior [62,63]; that is, the realization that COC behavior can be employed to successfully contain a TA herd. The presence of subsequent Hopf bifurcations therefore reflect the realization of this latent dynamic and its exploitation, as well as evidence that the appropriate control law has been learned.

Note that this interpretation of the nonlinear phenomena exhibited by human participants assumes that the transition between S&R and COC behavior is the result of a change in the parameter-dynamics that lead to a Hopf bifurcation. Accordingly, the model detailed above was designed to embody these potential dynamics (i.e., limit cycle or oscillator behavior). This assumption was based, in part, on previous research demonstrating that human arm movements exhibit spring-like properties consistent with Equations (4) and (7) [32,64] and the fact that human limbs (arms, legs) already entail the potential to produce both point attractive (discrete) and limit cycle (rhythmic) behavior (as detailed above). However, phase transitions due to a change in parameter-dynamics is not the only route to behavior change [65]. Instead, the discovery of COC behavior can also be modeled as a change in the underlying graph (systems of equations) used to define participant behavior. That is, it is possible that, instead of implementing task dynamics akin to Equation (7b) during S&R behavior, participants implement a more limited coordinative structure, defined by the following point attractor control:(9)$${\ddot{\theta }}_{i}=-\alpha_{\theta }{\dot{\theta }}_{i}-\omega_{\theta }^{2}\left(\theta_{i}-\theta_{T}\right).$$

If so, then the discovery of COC behavior reflected a change in the coordinative structure or system’s graph used to control the angular dynamics of participants’ movement (i.e., a transition from using Equation (9) to Equation (7b) by introducing the nonlinear escapement and coupling terms). Nonetheless, regardless of whether the discovery of COC behavior is best conceptualized as a change in parameter- or graph-dynamics, once it is discovered, its discovery still requires that participants learn to employ a control law similar to Equation (8) to induce an intentional Hopf bifurcation between S&R and COC behavior. Indeed, in either case, the Hopf bifurcation observed in the multiagent shepherding task appears to provide a signature characteristic of an intentional act [63], that, via the DPMPs used to model the shepherding behavior, provides key insights into how coordinative, synergistic rhythmic (limit cycle) and discreet (fixed-point) multiagent behavior can naturally emerge during ongoing task performance.

## 4. Conclusions

The aim of the current paper was to detail how a task-dynamic model generated from DPMPs can not only capture the behavioral modes observed in robust individual and multiagent behavior, but can reveal how the discovery and intentional enactment of effective, synergistic oscillator behavior can be spontaneously realized by a nonlinear transition process known as a Hopf bifurcation. This was achieved by reviewing recent research directed towards exploring and modeling the behavior dynamics observed during a dyadic, multiagent shepherding task. Although the task was designed as a laboratory game, it is important to appreciate that the encirclement of the TA herd afforded by COC behavior strikes a strong resemblance to the behaviors adopted by other animal systems that engage in shepherding or collaborative hunting. For example, sheepdog herd driving leads to the emergence of periods of oscillatory behavior where sheepdogs must drive the herd together while retrieving sheep that flee from the flanks of the herd [56]. During collaborative hunting, wolves and humpback whales, animals faced with a different set of constraints, converge on encirclement solutions to catch their prey. Wolves equally distribute themselves around a circle to isolate prey [66], while humpback whales create “bubble nets” which entrap fish [67]. Additionally, minimally cognitive artificial agents, when working within large groups, also converge on encirclement solutions to contain and transport agents [68]. Accordingly, the COC behavior observed by Nalepka and colleagues reflects a general shepherding strategy, one that is relatively invariant across task and multiagent contexts [61].

One should also appreciate that the DPMP based multiagent shepherding model summarized here provides just one example of how low-dimensional dynamics define synergistic, goal directed (multi) agent-environment behavior and that the same DPMPs can potentially be applied across countless multiagent task domains. However, although the model incorporates kinematic features of human limb movement, such as the amplitude–frequency relationship provided by the nonlinear escapement terms, the functions and model systems detailed here do not provide a generative account about how such low-dimensional structures emerge from lower-level biological components. In the literature, there are several models that seek to ground dynamical motor primitives with biological structures, such as central pattern generators (see [9] for a review of several models). Regardless, such functional/phenomenological models can still provide a way of uncovering the self-organizing principles that span and constrain particular neuromuscular substrates [33,69,70]. Additionally, the parameterization of such models via human movement timeseries may have relevance to medical diagnosis [71] and the determination of personality characteristics or preferences, such as risk-taking behavior [45].

Finally, the integration of model-based approaches to describing the features of human movements (i.e., the implementation of dynamical motor primitives) with model-free optimization approaches can provide a constrained flexibility in personalizing artificial agents embodying DPMP models for the use of human–machine interaction. In general, when it comes to learning complex behaviors, model-free techniques clearly work well in certain domains and theoretically can be guaranteed to converge upon an optimal behavior [72], but learning by trial and error is typically very slow, computationally expensive and fragile. Model-based techniques (i.e., Grey-box techniques), which are based on the low-dimensional dynamical primitives of an agent’s action or control system, not only operate to significantly reduce the action space that needs to be explored, but within the context of human action and interaction can lead to more human-like and responsive behavior, and therefore provide an advantage over model-free techniques. Furthermore, model-based solutions can more readily transfer between tasks with overlapping task dynamics, whereas model-free techniques usually require one to start completely from scratch when even the smallest property of the task dynamics is changed [73]. The challenge, however, is defining such rules for the agent. We believe DPMPs form a good foundation for a model-based approach for perceptual-motor behavior and, importantly, one that can be easily implemented within human–machine interaction systems. Moreover, due to the limited set of DPMPs that define human perceptual-motor behavior (i.e., discrete and rhythmic behaviors), they can be theoretically combined to form an *action grammar* [10] for complex actions with the assistance of model-free optimization techniques (e.g., reinforcement learning), which can be used for achieving flexible and on-the-fly re-parameterization and composition. Furthermore, multiple DPMPs can be combined together by assigning appropriate weights to them in order to generate even more complex trajectories for articulated models [74]. Indeed, adopting the latter, integrated, approach is likely to significantly advance the development artificial agents that are capable learning complex skills faster, while also being as flexible and robust as their human counterparts.

## Figures and Tables

**Figure 1 brainsci-10-00536-f001:**
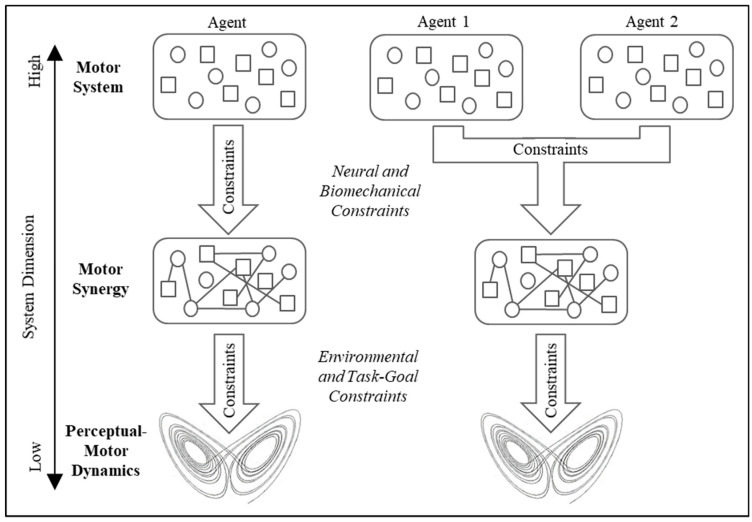
Illustration of the levels of constraint that result in synergy formation and task-specific perceptual-motor dynamics for both agent–environment and multiagent–environment systems. Adapted from Riley et al., 2011 [2]. See text for details.

**Figure 2 brainsci-10-00536-f002:**
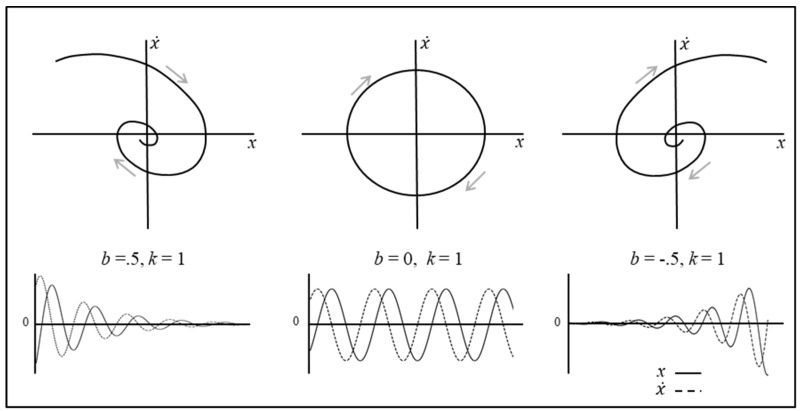
Illustration of the dynamics of the mass–spring system defined in Equation (1c). Three examples of how the behavior of the system evolves over time are plotted using (**top**) state space trajectories and (**bottom**) time-series graphs. See text for more details.

**Figure 3 brainsci-10-00536-f003:**
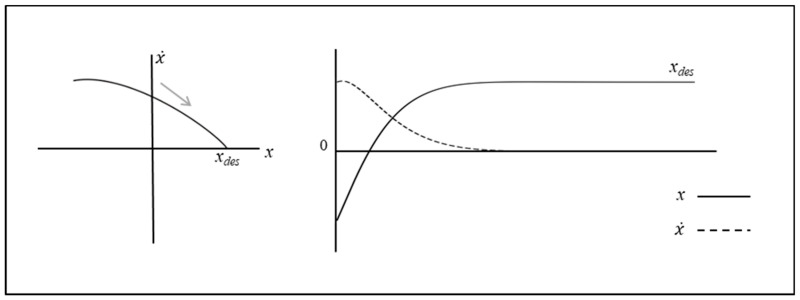
Illustration of the dynamics of the mass–spring system defined in Equation (2).

**Figure 4 brainsci-10-00536-f004:**
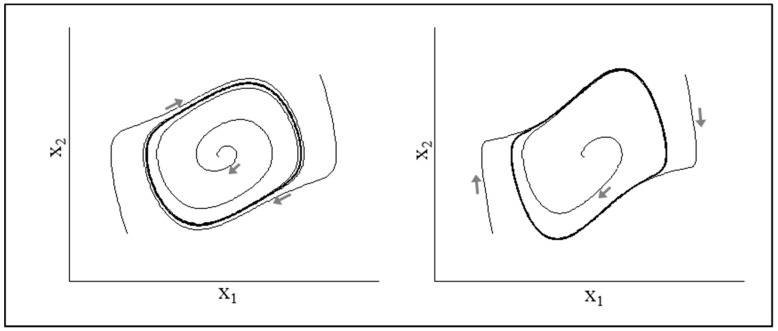
Example limit cycle trajectories of the van der Pol oscillator, with parameters settings of *b* and *k* = 1, and c = 0.5 (left) and c = 1 (right). The three trajectories shown in each graph correspond to initial conditions (−2.5, −2.5), (0.1, 0.1) and (2.5, 2.5). See text for more details.

**Figure 5 brainsci-10-00536-f005:**
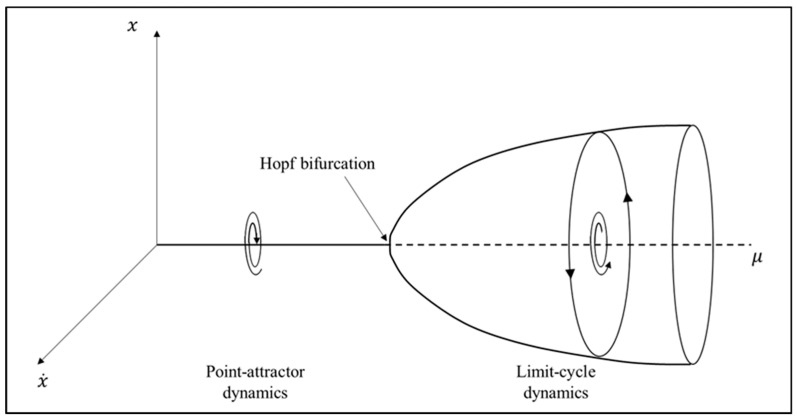
Fixed-point to limit cycle Hopf bifurcation diagram.

**Figure 6 brainsci-10-00536-f006:**
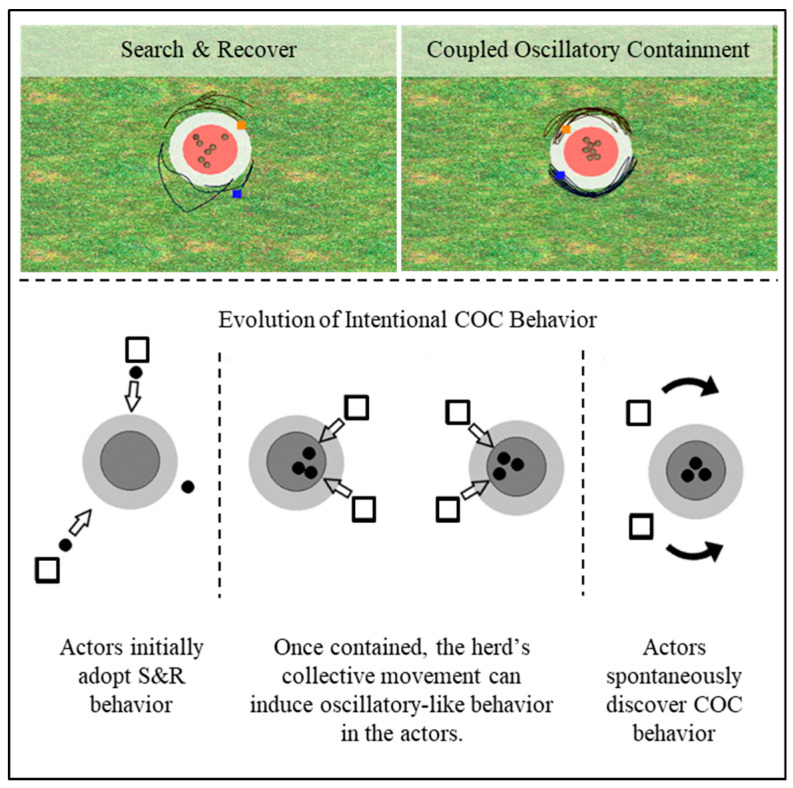
Shepherding behaviors and task space. (top) example of search and recover (S&R) and coupled oscillatory behavior (COC) (top right), (bottom) illustration depicting the interaction evolution that may scaffold COC discovery. See text for more details.

**Figure 7 brainsci-10-00536-f007:**
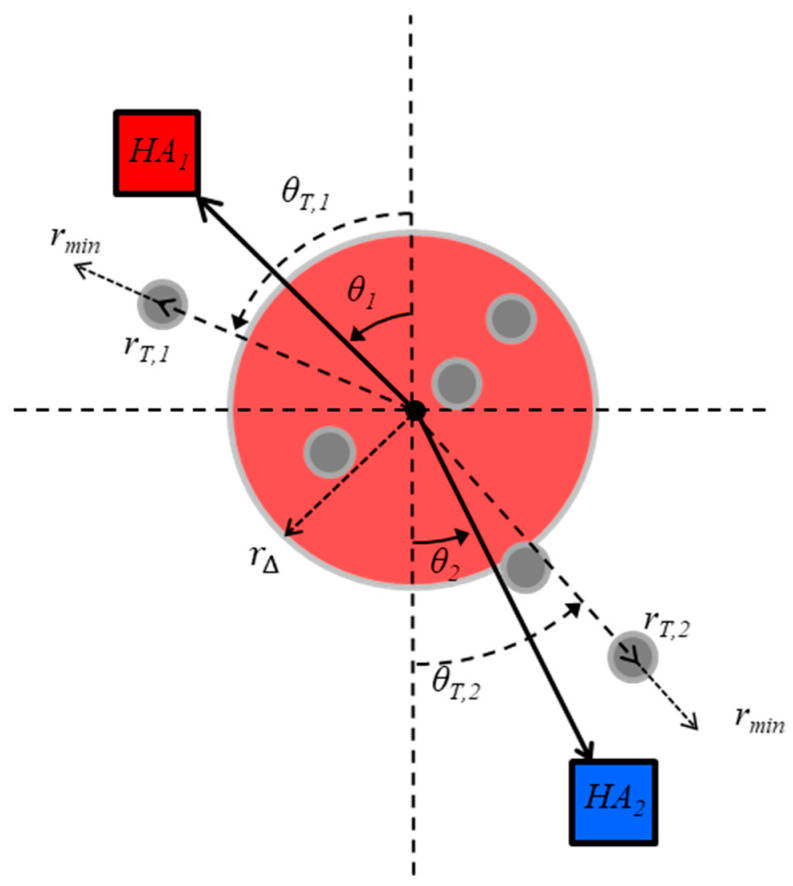
Illustration of the task space employed for the multiagent shepherding model. Herder-*i*’s of HA-*i*’s (where *i* = 1 or 2) location at any time within the game space is defined by a radial distance, $r_{i}$, and a polar angle, $\theta_{i}$. $r_{T,i}$ and $\theta_{T,i}$ correspond to the radial distance and polar angle, respectively, of the Target (TA) farthest from the origin on HA-*i*’s side of the game space. $r_{min}$ is HA-*i*’s minimum preferred radial distance for approaching the farthest TA, and $r_{\Delta }$ is the radius of the target containment area or the area within which the TAs are considered herded together. TAs with a radial distance greater than $r_{\Delta }$ are considered ‘uncontained’.

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
