# Peer review of "Hopf Bifurcations in Complex Multiagent Activity: The Signature of Discrete to Rhythmic Behavioral Transitions"

_brainsci, 2020, doi:10.3390/brainsci10080536_

Round 1
Reviewer 1 Report
In the revised version, the previous errors were corrected. I have no further comments. Please spellcheck before final submission.
Author Response
Thank you for your valuable feedback. We have spell-checked the entire manuscript and made corrections wherever necessary.
Reviewer 2 Report
In this article, the authors provide an overview of dynamical motor primitives and detail recent research demonstrating how these dynamical primitives can be used to model the task dynamics of complex multi agent behaviour. They review how a task-dynamic model of multi agent shepherding behaviour reveals how the discovery and intentional use of optimal behavioral coordination during task learning is marked by a spontaneous, self-organized transition between fixed-point and limit cycle dynamics (Andronov-Hopf bifurcation).
As I said in the first report, the topic is interesting but I do not think it is ready for publication in Brain Sciences, by the following reasons:
(i) the manuscript (as it is) is impossible to be read; there are some obscure parts that should be proved. The layout (combining old parts with reformulated ones) is very confusing for reviewers;
(ii) in Figure 1, there is a picture to Lorenz but it is not clear to me where it appears (the scheme is confusing);
(iii) Concerning the nonlinear Rayleigh number and the van der Pol term, no explanation is given about their interpretation. I do not understand the role of delta in (8).
(iv) the review is devoted to describe a list of models (as far as I know one comes from a lab game); they are simply models that may exhibit a limit cycle, a homoclinic cycle, a strange attractor....(wherever you want). But, the interesting thing would be to describe how to obtain this models in simple words and their evolution.
(v) Concerning the Hopf bifurcation, it is necessary to check the additional conditions. This is very important. Have you checked? Where are the computations?
In summary, the paper is not suitable for publication as it stands.
I recommend that the authors rewrite the paper taking into account my comments. If you want an expository and superficial paper, then I think that this journal is not suitable for this purpose. If you want a research article, things must be proven rigorously (if you do not want to exhibit all the details, move them to appendix).
I may give a third chance to the authors, but things should be written with a scientific rigor, without faults/mistakes/major errors. I do not recommend this paper (as it is) because it is full of unjustified steps.
Author Response
We have tried our best to address the reviewer's comments but some of the comments are very difficult to comprehend. This had been escalated to the editor and we were advised that a third reviewer will be invited to review the manuscript. We have addressed the comments from the third reviewer and we defer to the editor to make the final decision.
Reviewer 3 Report
This is my first review of this paper and I note that the authors have made many changes in response to comments of previous reviewers. I found the revised document to be very well written and generally grammatically correct. The statements are well supported with references.
As a general comment I would encourage the authors to ensure that assumptions aren't made with respect to readers' existing knowledge of esoteric terminology. Consider whether some terms need further explanation or definition to ensure that the messages in the paper are not lost on readers who are naive with respect to the terminology. As an example, the meaning of 'shepherding' and its relevance should be elaborated. This term seems to be quite central to the paper given its primacy in the statement of aim in the introduction. That being the case, it is confusing that the term is not mentioned explicitly in the abstract. Also, please check that the wording, emphasis and intent of the statements of aim in the Abstract, Introduction, and Conclusion are consistent.
Line 169-170: 'synergistic perceptual-motor has revealed..'. Is there a word missing after 'motor' e.g. 'behavior'?
Line 188. 'And, moreover...'. I understand the intent to add emphasis but nevertheless wonder whether both words are needed.
Line 338: Is this heading meant to be in italics (previously bold).
Author Response
As a general comment I would encourage the authors to ensure that assumptions aren't made with respect to readers' existing knowledge of esoteric terminology. Consider whether some terms need further explanation or definition to ensure that the messages in the paper are not lost on readers who are naive with respect to the terminology. As an example, the meaning of 'shepherding' and its relevance should be elaborated. This term seems to be quite central to the paper given its primacy in the statement of aim in the introduction. That being the case, it is confusing that the term is not mentioned explicitly in the abstract. Also, please check that the wording, emphasis and intent of the statements of aim in the Abstract, Introduction, and Conclusion are consistent.
We have added the necessary explanations and made sure the use of terms is consistent.
Line 169-170: 'synergistic perceptual-motor has revealed..'. Is there a word missing after 'motor' e.g. 'behavior'?
This has been corrected
Line 188. 'And, moreover...'. I understand the intent to add emphasis but nevertheless wonder whether both words are needed.
We agree that the intent is to add emphasis and we believe the sentence reads well with both the words
Line 338: Is this heading meant to be in italics (previously bold).
This has been corrected
This manuscript is a resubmission of an earlier submission. The following is a list of the peer review reports and author responses from that submission.
Round 1
Reviewer 1 Report
Summary:
The manuscript reviews dynamical movement primitives for the production of single- and multi-agent movements. Two types of systems are described: discrete movement generators, that have a start and an end point, and rhythmic generators for walking etc. Furthermore, two levels of constraints are introduced: task-level and biomechanics level. From the way the introduction is written, one might expect a focus on the biomechanics level in the rest of the review, but this is not the case: most of the application examples are about modeling on the task-level. Then, the authors give an overview over dynamical movement primitive models, and described dampened harmonic, van-der-Pol and other oscillators that have been used for movement modeling. They note that perception is necessary for planning and propose to include this into the DMP definition, which is certainly a good idea. They illustrate that depending on the dampening terms, these oscillators can be used for rhytmic behaviors (for undercritical dampening) or discrete ones (for overcritical dampening). As an application, they revisit their own work on sheperding ('An excellent example...' on line 293) by pairs of agents that are modeled by coupled oscillators and restate their previous finding that this yield behaviour which is convincing to human observers.
Suggestions:
- rewrite the introduction to make it clear that the biomechanics level is not the focus of the application examples in the rest of the review
- Multiagent behavior via oscillator coupling has been studied in depth before e.g. by (Mukovskiy and Giese) and others. Please broaden the scope of the review beyond the very narrow focus on the previous works by the authors.
- Most of the results seem to report on carefully hand-crafted models. One of the appealing features of DMPs is that they can be learned from data, or derived from an optimization procedure that minimizes some cost under task constraints. Please clarify why this has not been discussed in the text.
- In eqn. 3a, the dampening term does not have a minus sign (cf. eqn. 1a), yet in line 217 it is claimed that "..b>0...exhibit fixed-point dynamics...". Should that not be the other way round?
- In line 185, x_1 is introduced, but it presumably refers to x in eqn 3a ?
- eqn. 5: line 224:"It can be reformulated by combining..." please spell out this reformulation, or alternatively re-label the axes in figure 5. Right now this change of parameters seems to be motivated only by figure 5 which is not particularly well described.
- Eqn 9: "...which better resemble point attractor control...." why? Please explain
Author Response
- rewrite the introduction to make it clear that the biomechanics level is not the focus of the application examples in the rest of the review
- We have significantly shortened the introduction and better indicated that the focus of this article is on the second level of synergy formation portrayed in Figure 1. (i.e., the level of task/behavioral dynamics).
- Multiagent behavior via oscillator coupling has been studied in depth before e.g. by (Mukovskiy and Giese) and others. Please broaden the scope of the review beyond the very narrow focus on the previous works by the authors.
- See below
- Most of the results seem to report on carefully hand-crafted models. One of the appealing features of DMPs is that they can be learned from data, or derived from an optimization procedure that minimizes some cost under task constraints. Please clarify why this has not been discussed in the text.
- In conjunction with comment 2, we extended our discussion in the conclusion to illustrate this very point. Our hand-crafted model is to demonstrate the concept of Hopf bifurcations in a multiagent task setting.
- In eqn. 3a, the dampening term does not have a minus sign (cf. eqn. 1a), yet in line 217 it is claimed that "..b>0...exhibit fixed-point dynamics...". Should that not be the other way round?
- This has been corrected.
- In line 185, x_1 is introduced, but it presumably refers to x in eqn 3a ?
- This has been corrected
- eqn. 5: line 224:"It can be reformulated by combining..." please spell out this reformulation, or alternatively re-label the axes in figure 5. Right now this change of parameters seems to be motivated only by figure 5 which is not particularly well described.
- We have modified the equation and the description to correct this
- Eqn 9: "...which better resemble point attractor control...." why? Please explain
- The sentences surrounding Eq. 9 have been edited to state explicitly that Eq. 9 exhibits point attractive dynamics.
Reviewer 2 Report
Report on the paper:"Hopf Bifurcations in Complex Multiagent Activity: The Signature of Discrete to Rhythmic Behavioural Transitions"
In the paper under review, the authors give an overview of dynamical motor primitives and detail recent research demonstrating how these dynamical primitives can be used to model complex multiagent behaviour. They explain how a task-dynamic model of multiagent shepherding behavior, compose of rudimentary fixed-point and limit cycle dynamical primitives, can not only effectively model the behaviour of cooperating human co-actors, but also reveals how the discovery and intentional use of optimal behavioural coordination during task learning is marked by a spontaneous, self-organised transition between fixed-point and limit cycle dynamics thorugh a Hopf bifurcation.
The topic is interesting but it would be a good idea for the author to decide what the main contribution/result of the paper is and to state that more clearly. The paper takes quite some pages before getting to the point; my feeling is that a shorter paper without the extensive descriptions of existing theory, would be more appreciated by experts.
As it is now, all this information obscures the novel ideas and results that do exist in the paper.
A revised version of the paper has its merits, but given the problems
with the paper I don't see a compelling case to publish in Brain Sciences.
Additional problems:
(i) In Maths, a discrete system cannot be modelled by a differential equation (I guess you mean different things, but be care with words).
(ii) Table 1 is very confusing. It should be rewritten.
(iii) How van der pol Equation can be adapted to your work?
(iv) Derivation of equation (6) should be done. The same for others.
(v) Conclusion should be shorter.
(vi) Do you really need 77 references?
Author Response
- The topic is interesting but it would be a good idea for the author to decide what the main contribution/result of the paper is and to state that more clearly. The paper takes quite some pages before getting to the point; my feeling is that a shorter paper without the extensive descriptions of existing theory, would be more appreciated by experts.
- We have significantly edited the introduction to better indicate the aim of the manuscript. We have left the tutorial sections on dynamical systems as to make the manuscript accessible to non-experts.
- In Maths, a discrete system cannot be modelled by a differential equation (I guess you mean different things, but be care with words).
- We define what is a discrete behavior in the human movement sciences in the introduction. This is the definition used throughout the manuscript when discussing discrete behaviors.
- Table 1 is very confusing. It should be rewritten.
- We have removed Table 1.
- How van der pol Equation can be adapted to your work?
- We use the van der Pol oscillator as an example of a system exhibiting limit-cycle dynamics. The shepherding model we present incorporates van der Pol as well as Rayleigh nonlinear escapement terms to produce the limit-cycle dynamics observed in human data.
- Derivation of equation (6) should be done. The same for others.
- We disagree. This is necessary for this paper meant to be accessible to non-experts and to a wider audience, including researchers in the human-movement, psychological, cognitive sciences.
- Conclusion should be shorter.
- We believe three paragraphs is sufficient to relate the shepherding dynamics exhibited by humans to other systems, a discussion of limitations of the modelling approach, as well as applications.
- Do you really need 77 references?
- We have attempted to be as comprehensive as possible with regards to citing previous research and we defer to the Editor to make the determination whether the number of references need to be shortened for publication.